# Influence of malaria, soil-transmitted helminths and malnutrition on haemoglobin level among school-aged children in Muyuka, Southwest Cameroon: A cross-sectional study on outcomes

Irene Ule Ngole Sumbele[1,2]*, Ayeah Joy Nkain[1¤], Teh Rene Ning[1], Judith Kuoh Anchang-Kimbi[1], Helen Kuokuo Kimbi[1,3]

1 Department of Zoology and Animal Physiology, University of Buea, Buea, Cameroon, 2 Department of Microbiology and Immunology, Cornell College of Veterinary Medicine, Ithaca, New York, United States of America, 3 Department of Medical Laboratory Science, The University of Bamenda, Bamenda, Cameroon

¤ Current address: Department of Gynaecology and obstetrics, University of Ibadan, Ibadan, Nigeria
* sumbelei@yahoo.co.uk

**Data Availability Statement:** All relevant data are within the manuscript.

## Abstract

### Background

The health of school-aged children (SAC) is often compromised by malaria parasitaemia (MP), soil-transmitted helminths (STH), and malnutrition in the tropics. The aim of this study was to determine the prevalence and influence of MP, STH and malnutrition on haemoglobin (Hb) levels as well as identify its predictors.

### Methods

This cross-sectional study was carried out in SAC (4–14 years) in Owe, Mpundu and Meanja villages in Muyuka, Southwest Cameroon. Hb concentration was measured using a URIT-12 Hb meter while MP and STH were determined by Giemsa staining of blood films and Kato-Katz technique respectively. Anthropometric measures (weight, height and mid upper arm circumference (MUAC)) of malnutrition (z-scores of <−2 standard deviations below mean) were obtained by standard methods. Categorical and continuous variables were compared appropriately, and multiple linear regression model was used to determine predictors of Hb level.

### Results

The prevalence of MP, STH, anaemia and malnutrition in the 401 SAC examined were 33.9%, 2.2%, 75.3% and 24.4% respectively. The prevalence of MP varied significantly with locality (P = 0.031). Stunting occurred commonly (23.7%) and was significantly higher in males (28.6%), children 11–14 years old (38.3%) and those of Meanja locality (47.4%) than their counterparts. Significantly higher prevalence of anaemia was observed in children of Meanja (89.5%) and those both MP positive and malnourished (86.2%). Moderate anaemia occurred commonly (60.6%) and children ≤6 years old had significantly (P = 0.034) higher

**Funding:** The author(s) received no specific funding for this work.

**Competing interests:** The authors have declared that no competing interests exist.

prevalence (75.0%). Mean Hb level varied significantly (P = 0.004) with age and those ≤6 years old infected with MP had significantly (P = 0.022) lower values. Significant predictors of Hb levels were the MUAC (P <0.001) and the MP status (P = 0.035). Based on the Hb level (>11g/dL) and the absence of MP, STH and malnutrition, 13.7% of the SAC were considered as healthy.

## Conclusions

The health of a majority of SAC is compromised by malaria, helminthiasis, malnutrition and other conditions not investigated. Anaemia is of major public health concern hence, intervention programmes that integrate malaria control with improvement of educational levels especially on proper nutrition and health care practices are desirable.

## Background

Health, as defined by the World Health Organization (WHO), is a state of complete physical, mental and social well-being and not merely the absence of disease or infirmity [1]. The relative level of wellness and illness of an individual which considers the presence of biological or physiological dysfunction, symptoms and functional impairment defines the health status of an individual. The health of school-age children (SAC) is usually compromised by common diseases such as malaria (which remains a considerable public health problem in much of the tropics and subtropics) and helminthiasis [2–4]. In Cameroon, malaria and soil-transmitted helminth (STH) infections are both widespread and are accountable for increased morbidities and associated consequences in vulnerable populations, including SAC [5–8].

*Plasmodium falciparum* is the most prevalent malaria parasite in the WHO African Region, accounting for 99.7% of estimated malaria cases in 2017 and Cameroon is amongst the 11 high—burden countries that account for more than 70% of the global malaria cases and deaths [2]. *P. falciparum* is also the most pathogenic species and remains a major cause of morbidity and mortality, with children less than five years of age and pregnant women severely affected [9]. In 2017, children aged under 5 years accounted for 61% (266 000) of all malaria deaths worldwide [2]. Consequently, control measures in endemic regions have focused on the protection of these two groups at highest risk of malaria disease. On the other hand, older children who are less often symptomatic, and may play an important role in transmission, have not traditionally been the focus of intensive detection and control strategies [10].

School age children rather than preschool children or adults, are most at risk of *Plasmodium* helminth co-infection and thereby at greatest risk of the consequences of co-infection [3]. Co-infections with helminth and malaria parasites have negative impact upon host and synergisms between multiple parasite species infections and infection intensity are known to exacerbate anaemia [3, 8, 11]. STH infections can accelerate or exacerbate malnutrition hence infections with STH and malaria parasite could singly or combined be contributing factors of malnutrition and/or anaemia as shown by several studies [12–15].

Malnutrition is said to be the underlying cause of deaths in 48% of children below 5 years in Cameroon [16] while the burden of malnutrition in SAC is infrequently determined especially in rural areas. Common nutritional indicators of subclinical undernutrition such as underweight, wasting and stunting in children are proxy indicators of overall well-being and reflect, the burden of infectious diseases in the community [17]. The nutritional status of SAC

impacts their health, cognition, and subsequently their educational achievement [18]. Manifestation of malnutrition is often observed in terms of anaemia, micronutrient deficiencies (iron, folic acid, riboflavin, vitamin A and $B_{12)}$ and anthropometric measurements. Hence, the assessment of nutritional status of this population segment is essential for making progress towards improving the overall health of SAC [19].

Hb concentration is the most reliable indicator of anaemia at the population level, as opposed to clinical measures which are subjective and therefore have more room for error [20]. Anaemia is an indicator of both poor nutrition (micronutrient deficiency) and poor health (chronic infections, predominantly malaria, hereditary haemoglobinopathies). In most developing countries, anaemia is a public health problem and SAC are more vulnerable due to their rapid physical and physiological development [20, 21]. The prevalence of anaemia among SAC in Africa ranges from 64.3% to 71% [20] and in the Mount Cameroon area it ranges from 19.8% to 44.2% [22, 23]. Anaemia is considered as a public health problem (prevalence $\geq$ 5%) when the Hb value is below the population specific Hb threshold. The prevalence of anaemia is objective and quantifiable and can be measured in the most remote areas where access to health care is a challenge. In addition, anaemia is a major complication of several neglected tropical diseases (NTD). Moreover, it changes in predictable fashions with alterations in disease burden [24].

While studies may have been carried out on single infections with malaria parasite or co-infections with STHs in different populations, there is a dearth of knowledge on the health status of SAC in Cameroon especially in rural settings. Thus, the main objectives of this study were to determine the prevalence of MP, STH and malnutrition, assess their influence on Hb levels as well as identify the predictors of Hb levels. We hypothesised that the level of haemoglobin is a valid indicator in predicting the health status of SAC in rural areas endemic for malaria and NTDs such as soil-transmitted helminths.

## Methods

### Study sites

The study sites comprised of Owe, Mpundu and Meanja villages of Muyuka Sub-division located at the foot of Mount Cameroon. Owe is located at an altitude of 59m, latitude 4˚17'23" N, longitude 9˚ 22'50" E to 84m a.s.l., latitude 4˚18'00" N and longitude 9˚ 22'32" E. Mpundu is located at an altitude of 54m, latitude 4˚14'14" N, longitude 9˚24'44" E, to 68m a.s.l, latitude, 4˚ 14'42" N and longitude 9˚23'40" E. Meanja is located at 62m above sea level (a.s.l.), latitude 4˚ 14'53"N, longitude 9˚23'48" E to 69m a.s.l., latitude 4˚ 15'53"N and longitude 9˚24'48" E as shown in Fig 1. Owe, Mpundu and Meanja villages are 7km, 12km and 5.6 km away from Muyuka town respectively. The rainy season in the area, runs from March to October while the dry season runs from November to mid-March. Annual temperature ranges from 18–35˚C. The natives of these villages are known as the Balongs [25]. Subsistence and small-scale cash crop farming constitute the mainstay of the villages. Other nearby landmarks include: River Mungo, a Cameroon Development Corporation (CDC) workers' camp and CDC palm and rubber plantations. Owe has no closed potable water source but has streams which serve as a source of drinking water and other household activities while Meanja and Mpundu have closed potable water source. They lack health facilities such as clinics/hospitals and residents tend to visit pro-pharmacies more often when sick than hospitals for drugs.

### Study design

The cross-sectional study was carried out simultaneously in three localities between the months of March and June 2015.

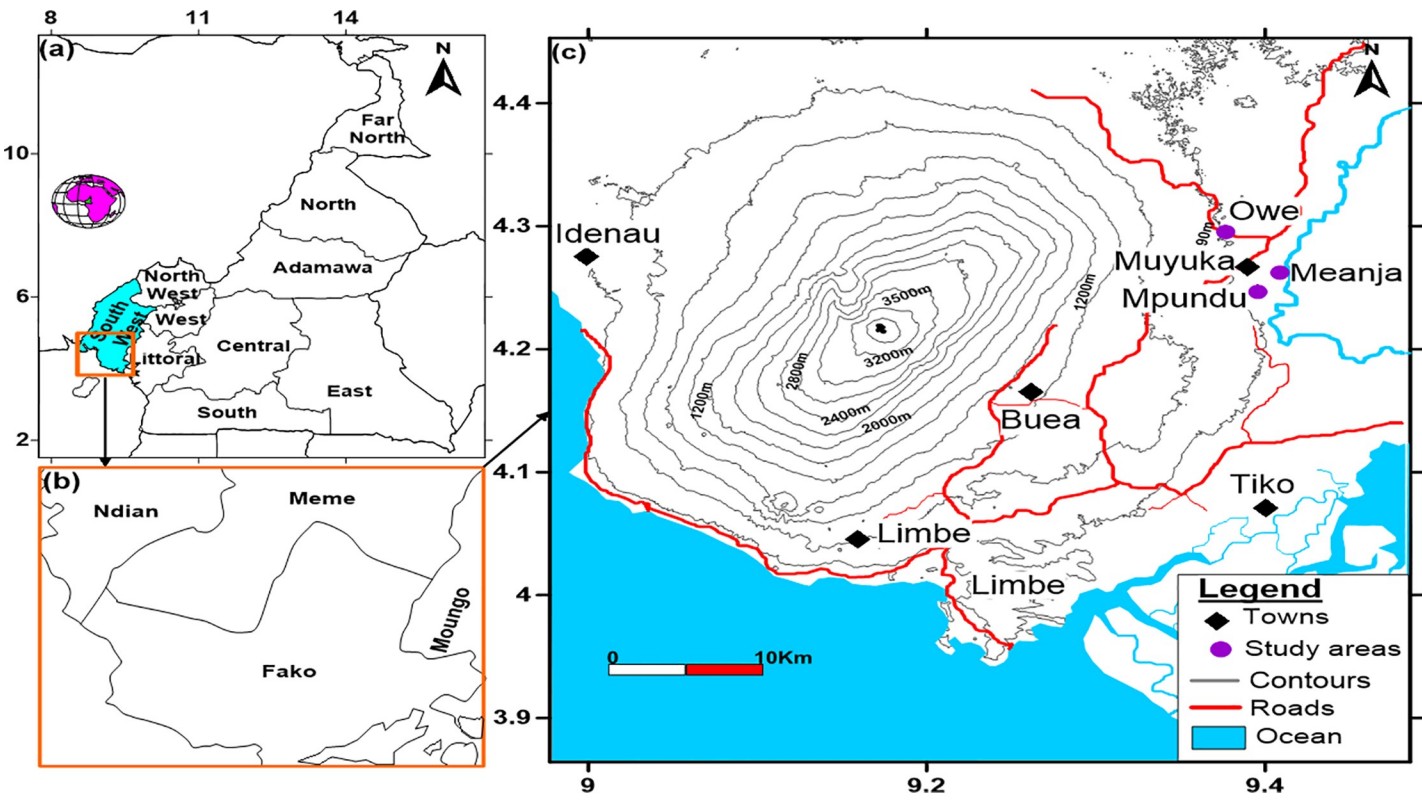

**Fig 1. Map showing the location of study areas in Muyuka, Southwest Cameroon.**

**Study population.** The study population constituted SAC (4–14 years old) of both sexes whose parents or legal representatives had signed a written consent/assent form. Primary schools were selected at random from a list of schools operating in these villages and children selected at random from each class.

**Sampling size.** The sample size was determined using the formula n = $Z^2pq/d^2$ [26], where n represented the sample size required; Z was 1.96, which is the standard normal deviate (for a 95% confidence interval, CI); p was 35.5%, the prevalence of *P. falciparum* [8] or 14.0%, the proportion of helminth infection [27]; q was 1-p, the proportion of malaria parasite negatives or helminth negative; and d was 0.05, the acceptable error willing to be committed. A minimum sample size of 268 was obtained from the average of the calculated sample size for the prevalence of *P. falciparum* (351) and that of helminth (185).

**Sampling procedure.** Five (05) schools (Government school (GS) Owe, Catholic school (CS) Owe, GS Mpundu, CS Mpundu and CS Meanja) selected by random sampling from a list of nine schools operating in these villages agreed to take part in the study. A representative number of children were selected at random by balloting from each class. First, the children, head teachers, and village chiefs were sensitized on the protocol and benefits of the exercise to the children and to the community. Informed consent forms were sent to parents/guardians through the pupils explaining the purpose and benefits of the study as well as the precautions taken to minimize risk. Children who returned with signed consent forms were enrolled in the study.

The samples collected comprised of finger prick blood for MP detection and speciation, and Hb measurement for assessment of anaemia as well as stool sample for the detection of egg or larva of STHs. The investigative methods included the use of questionnaires, clinical

evaluation and laboratory methods such as Giemsa-stained microscopy and Kato-Katz technique.

## Questionnaire

A simple pre-tested structured questionnaire (S1 File) was administered to each pupil with the assistance of schoolteachers and investigators. Information on important demographic data such as age, sex, area of residence was obtained as well as information relating to malaria and STH infection epidemiology. In brief, the questionnaire sought to obtain information on insecticide treated net (ITN) possession and use, house type, bushes around the residence, pre-fever history, history of anti-malarials, toilet availability and type, hand washing, shoe wearing, and farming frequency.

## Clinical evaluation

The axial temperature of each child was measured using a mercury thermometer to determine the presence of fever. Children who had temperatures $\geq 37.5°C$ were reported to have fever.

The age, height, weight and mid upper arm circumference (MUAC) of participants were measured in order to determine anthropometric indices. Height and weight measurements were recorded to the nearest 0.1cm and 0.5kg respectively. Weight was measured using a mechanical weighing scale and heights and MUAC were measured using a measuring tape. Ages of children were obtained from school records with the permission of the head teacher. The z scores height-for-age, weight-for-age, weight-for-height were computed based on the WHO 2006 growth reference curves [28]. Stunting was defined as height-for-age z (HAZ) score of < -2; underweight as weight-for-age z (WAZ) score of < -2 and wasting as weight-for-height z (WHZ) score < -2. A child was considered malnourished if he or she scored < -2 in one of the z scores (HAZ, WAZ, WHZ). Z score of < -3 indicated severe stunting, under-weight and wasting.

## Collection of blood and stool samples

Finger prick blood was collected from each pupil with signed assent form. The first drop of blood was immediately wiped off and the next drop was placed on the Hb test strip already inserted into the URIT-12 haemoglobin meter (URIT Medical Electronic co., Ltd, London, United Kingdom) for Hb determination. The Hb value displayed was recorded to the nearest 0.1 g/dL. A participant was considered anaemic when Hb concentration fell below the WHO reference values for age or gender [29]. Anaemia was further classified as either mild (between 10.0 g/dL and the level in the WHO reference values for age), moderate (7.0–9.9 g/dL based on WHO reference for age) or severe (<7 g/dL and WHO reference for age) [29]. Two drops of whole blood were placed on a labelled, grease-free glass slide 1mm apart for the preparation of thin and thick films for the detection and speciation of malaria parasites, as described by Cheesbrough [30].

Each child was provided with a labelled screw-capped plastic container to return with a fresh midday stool sample. A single stool sample was collected from each participant. The fresh samples were preserved with 10% formalin to maintain the morphology of the egg. Samples were transported in a specimen box to the Malaria Research Laboratory of the University of Buea for diagnosis by the Kato-Katz technique.

## Laboratory procedures

Both thin and thick blood films prepared in the field were air-dried. The thin blood film was fixed in 75% methanol, and both thick and thin blood films were stained using 10% Giemsa

solution for 15 minutes [30]. Slides were then microscopically examined for the presence of malaria parasites by two independent parasitologists, and in the case of any disparity they were read by a third parasitologist. Slides were considered positive when asexual forms and/or gametocytes of any *Plasmodium* species were observed on the blood film. Parasite density per μL of blood was determined based on the number of parasites per 200 leukocytes on thick blood film assuming a white blood cell count (WBC) of 8000 leucocytes/uL of blood [30]. Asymptomatic malaria parasitaemia was defined as the presence of *Plasmodium* parasites with an axillary temperature of $< 37.5°C$, while clinical malaria parasitaemia was defined as the presence of any species of *Plasmodium* together with an axillary temperature of $\geq 37.5°C$ or reported fever in the previous 48 hours, or headache or joint pain. Parasitaemia was classified as low ($\leq 500$ parasite /μL of blood), moderate (501–5000 parasites/μL of blood) and high ($>5000$ parasites/μL of blood).

Stool smears were prepared and examined using the Kato-Katz thick smear method, as described by Cheesbrough [30]. Duplicate smears were prepared for each specimen. Each slide was allowed to clear for 30 min, and then examined at 100× total magnification within one hour of preparation to avoid missing hookworm eggs. Morphological identification of eggs of *A. lumbricoides*, *T. trichiura* and hookworm were based on identification aids [30]. All the eggs in the 41.7 mg of stool were counted and multiplied by 24 to compute the number of eggs per gram of faeces (epg). As a quality control measure, all positive slides and 20% of randomly selected negative smears were re-examined within 24 hours by a third experienced parasitologist who had no knowledge of the previous results. An average of the counts was utilised.

## Statistical analysis

All data collected was entered into Microsoft Excel and cleaned for entry errors (MS Excel 2016). Data was further analysed with the IBM-statistical package for the social sciences (IBM-SPSS) version 19 (IBM—SPSS, Inc, Chicago, IL, USA). Data was summarized into means and standard deviations (SD), and proportions used in the evaluation of descriptive statistics. Prevalence of malaria, STH infections, anaemia and malnutrition were determined and compared using the Chi-square ($\chi^2$) test. The mean (SD) Hb levels were compared using non-parametric tests [(Mann Whitney U and Kruskal Wallis tests) and parametric tests (t-test and analysis of variance (ANOVA)] where appropriate. Malaria parasite counts were log transformed before analysis. The Pearson correlation coefficient (r) was used to evaluate the linear correlation between age, sex, MUAC, nutritional status, MP status, level of education, type of school, locality and Hb level. The attributable risk (AR%) of anaemia caused by malaria and malnutrition was calculated accordingly [31]: $[(n_1 m_0 - n_0 m_1)/n(n_0+m_0)] \times 100$, where $n_0 =$ anaemic children without malaria and $n_1 =$ anaemic children with malaria, whereby $n_0 + n_1 = n$, $m_0 =$ non anaemic children without malaria, and $m_1 =$ non anaemic children with malaria, whereby $m_0 + m_1 = m$. The multiple linear regression (MLR) model (enter) with Hb as the dependent variable was run to examine the influence of the following independent variables; age, sex, MUAC, nutritional status, MP status, level of education, type of school and locality. All 401 participants were included in the model. Significant levels were measured at 95% confidence interval (CI) with significant differences set at $P < 0.05$.

## Ethics statement

Administrative clearances were obtained from the Regional Delegation of Basic Education as well as from the Catholic Education Board. The institutional review board hosted by the Faculty of Health Sciences, University of Buea issued the ethical clearance document after reviewing the study protocol, participant's information sheet and assent forms. Authorization to

proceed with the study in these villages within the selected school was obtained from village chiefs and school head teachers. Children participated in the study if a parent or guardian signed the informed consent form. The parents or guardian and their children were informed that their participation in the study was voluntary and they could withdraw at any time without any explanation.

## Results

### Socio-demographic characteristics of the study participants

A total of 401 pupils with a mean (SD) age of 8.94 (2.9) years (range = 4–14) of both sexes (212 females and 189 males) were examined. Majority of the participants were pupils from Owe locality (47.6%), Catholic schools (58.1%) and between the age range of 7–10 years (58.4%). A greater proportion of the participants used bed nets (75.1%) and lived in plank houses (85%) as shown in Table 1. The proportion of children whose house floors were cemented (93.0%) was higher compared to their counterparts with earthen house floors (0.7%).

### Clinical profile of participants

The mean (SD) HAZ, WAZ, and WHZ scores were -1.17 (1.2), -0.34 (1.2) and 2.91 (4.0) respectively. The prevalence of fever, MP, STH infections, anaemia and malnutrition were 15% (95% CI = 11.7–18.7%), 33.9% (95% CI = 29.5–38.7%), 2.2% (95% CI = 1.8–2.7%), 75.3% (95% CI = 70.9–79.3%) and 24.4% (95% CI = 20.5–28.9%) respectively (Table 2).

All MP detected were *Plasmodium falciparum*. Out of the 9 children with STH infection, 5 (55.6%) were infected with *Trichuris* and 4 (44.4%) with *Ascaris*, no mixed infections, hookworm, and *Strongyloides* were observed. Out of the 60 (15%, 95% CI = 11.7–18.7%) pupils with fever, 22 (36.7%, 95% CI = 25.2–49.4%) of them were positive for MP. ITN usage was highest in children ≤ 6 years old (72.1%), followed by 7–10 years (64.8%) and least in 11–14 years old (53.3%).

### MP, STH and malnutrition prevalence

Out of the 136 children with malaria parasitaemia 114 (83.8%, 95% CI = 76.7–89.1%) of the infections were asymptomatic. MP prevalence varied significantly (P = 0.031) among the

**Table 1. Characteristics of the study participants.**

| Characteristic | | Number examined | % |
|---|---|---|---|
| **Sex** | Male | 189 | 47.1 |
| | Female | 212 | 52.9 |
| **Age group (years)** | ≤ 6 | 60 | 15.0 |
| | 7–10 | 234 | 58.3 |
| | 11–14 | 107 | 26.7 |
| **Locality** | Owe | 191 | 47.6 |
| | Meanja | 38 | 9.5 |
| | Mpundu | 172 | 42.9 |
| **Schools** | Government school | 168 | 41.9 |
| | Catholic school | 233 | 58.1 |
| **ITN use** | Yes | 301 | 75.1 |
| | No | 100 | 24.9 |
| **House type** | Plank | 341 | 85.0 |
| | Block | 60 | 15.0 |

**Table 2. Clinical characteristics of the 401 SAC.**

| Characteristics | Number | % |
|---|---|---|
| Prevalence | | |
| Prevalence of fever ($\geq$ 37.5˚C) | 60 | 15.0 |
| Overall prevalence of MP | 136 | 33.9 |
| Overall prevalence of STH | 9 | 2.2 |
| Prevalence of anaemia | 302 | 75.3 |
| Prevalence of malnutrition | 98 | 24.4 |
| Means | Mean (SD) | Range |
| Mean age in years | 8.9 (0.1) | 4–14 |
| Mean weight in kg | 124.6 (0.6) | 98–157 |
| Mean height in cm | 27.29 (0.3) | 15–53 |
| Mean MUAC in cm | 18.6 (2.08) | 13.6–33.0 |
| Mean HAZ | -1.20 (1.2) | -5.50–2.40 |
| Mean WAZ | - 0.35 (1.2) | -2.76–5.51 |
| Mean WHZ | 0.93 (1.6) | -2.79–8.28 |
| Mean temperature in˚C | 36.85 (0.1) | 33–39.5 |
| Mean haemoglobin in g/dL | 10.6 (1.4) | 4.4–14.3 |
| Mean MP density | 1326.2 (3939) | 40–34000 |

HAZ = Height-for-age z score; MP = Malaria parasite; MUAC = Mid upper arm circumference; STH = Soil-transmitted helminths; WAZ = Weight-for-age z score; WHZ = Weight-for-height z score.

localities only. The children of Meanja had the highest prevalence of MP (50.0%) when compared with their Owe and Mpundu equivalent (Table 3).

STH was prevalent in 2.2% (95% CI = 1.8–2.7%) of the children with no significant differences observed with respect to gender (P = 0.619), age (P = 0.139), locality (P = 0.431), and the type of school (P = 0.500) as shown in Table 3.

Of the 98 children with malnutrition, stunting the most common form occurred in 95 (96.9%), while underweight and wasting occurred in 18 (18.4%) and 1 (1.0%) of them, respectively. Overall, the prevalence of stunting was 23.7% (95% CI = 19.8–28.1%), underweight was 4.5% (95% CI = 2.9–7.0%) while wasting was 0.3% (95% CI = (0.06–1.8%). The prevalence of malnutrition, more specifically stunting was significantly higher in males (28.6%), children of the 11–14 years age group (38.3%) and those of Meanja locality (47.4%) than their respective counterparts as shown in Table 3.

## Anaemia prevalence and its severity

Overall, anaemia occurred in 75.3% (95% CI = 90.9–79.3%) of the pupils with similar prevalence in males (74.6%) and females (75.9%). Although not statistically significant, the prevalence of anaemia was higher in the $\leq$ 6 years age group (81.7%), children from GS (78.6%) and those positive for MP (80.9%) than their contemporaries. However, significantly higher (P = 0.035) prevalence of anaemia was observed in children of the Meanja locality (89.5%) when compared with those of Owe (70.7%) and Mpundu (77.3%) as revealed in Table 4.

Moderate anaemia (Hb = 7.0–9.9 g/dL) occurred commonly (60.6%, 95% CI = 55.7–65.3%) in the participants than mild (13.0%, 95% CI = 10.3–16.6%) and severe anaemia (1.8%, 95% CI = 0.9–3.6%). While no significant differences in anaemia severity were observed by sex, locality, type of school, MP status and density, the prevalence of moderate anaemia was highest in children of the $\leq$ 6 years age group (75.0%) and the difference was statistically significant at P = 0.034 (Table 4).

**Table 3. Prevalence of MP, STH and malnutrition by sex, age, locality and type of school.**

| Parameter | Category | N | MP % (n) | STH % (n) | Stunting % (n) | Underweight % (n) | Wasting % (n) | Malnutrition % (n) |
|---|---|---|---|---|---|---|---|---|
| **Sex** | Male | 189 | 34.9 (66) | 2.6 (5) | 28.6 (54) | 4.8 (9) | 0.0 (0) | 29.6 (56) |
| | Female | 212 | 33.0 (70) | 1.9 (4) | 19.3 (41) | 4.2 (9) | 0.7 (1) | 19.8 (42) |
| | P- value | | 0.688 | 0.619 | **0.03** | 0.803 | 0.304 | **0.022** |
| **Age group in years** | ≤ 6 | 60 | 36.7 (22) | 1.6 (1) | 6.7 (4) | 1.7 (1) | 1.7 (1) | 10.0 (6) |
| | 7–10 | 234 | 29.5 (69) | 1.3 (3) | 21.4 (234) | 3.4 (8) | 0.0 (0) | 21.8 (51) |
| | 11–14 | 107 | 42.1 (45) | 4.7 (5) | 38.3 (41) | 8.4 (9) | 0.0 (0) | 38.3 (41) |
| | P–value | | 0.067 | 0.139 | **< 0.001** | 0.061 | 0.117 | **< 0.001** |
| **Locality of School** | Owe | 191 | 28.8 (55) | 2.1 (4) | 25.1 (48) | 6.8 (13) | 0.7 (1) | 26.7 (51) |
| | Meanja | 38 | 50.0 (19) | 5.1 (2) | 47.4 (18) | 2.6 (1) | 0.0 (0) | 47.4 (18) |
| | Mpundu | 172 | 36.0 (62) | 1.8 (3) | 16.9 (29) | 2.3 (4) | 0.0 (0) | 16.9 (29) |
| | P–value | | **0.031** | 0.431 | **<0.001** | 0.101 | 0.560 | **<0.001** |
| **School Type** | Government | 168 | 30.4 (51) | 3.0 (5) | 21.4 (36) | 6.0 (10) | 0 0 (0) | 22.0 (37) |
| | Catholic | 233 | 36.5 (85) | 1.7 (4) | 25.3 (59) | 3.4 (8) | 0.6 (1) | 26.2 (61) |
| | P- value | | 0.201 | 0.500 | 0.366 | 0.229 | 0.367 | 0.339 |

P- value obtained using $\chi^2$. P -values in bold are statistically significant.

## Infection status, anaemia prevalence and haemoglobin levels

Of the 401 participants, 48.6% (195) had no MP, STH or malnutrition. Correspondingly, 25.7% (103) had MP only, 0.7% (3) had STH only,16.2% (65) were malnourished only, 7.2% (29) had both MP and were malnourished, 0.5% (2) were infected with STH and malnourished and 1.0% (4) were co-infected with MP and STH. The prevalence of anaemia and mean (SD) Hb levels in g/dL as influenced by infection category and nutritional status is shown in Table 5. Children who were MP positive and malnourished as well, had the highest prevalence of anaemia (86.2%) followed by those infected with MP only (79.6%) and malnutrition only (79.6%). However, the AR of anaemia caused by MP + malnutrition, MP only and malnutrition only were 0.96%, 3.8%, and 1.1% respectively.

Overall, the mean (SD) Hb varied significantly (P = 0.004) with age with the highest level occurring in children 11–14 years old [10.9 (1.5) g/dL] and the lowest in the ≤ 6 years old [10.2 (1.2) g/dL]. As shown in Table 5, the mean (SD) Hb (g/dL) levels in all conditions were lower than the normal Hb levels for age and sex. Statistically significant (P = 0.022) only, was the difference in mean Hb levels with age in children infected with MP where, children ≤ 6 years old had the lowest mean Hb [10.0(0.9) g/dL] when compared with the 7–10 years [10.4 (1.2) g/dL] and the 11–14 years [10.9 (1.2) g/dL] age groups. The mean Hb level was lowest in those with parasitaemia of 15000 parasites/μL when compared with the aparasitaemic and the other quartiles of malaria parasite densities as shown in Fig 2.

## Predictors of haemoglobin levels

The tolerance statistics of the multiple linear regression (MLR) model were all below 1 and all the variance inflation factors (VIF) were less than 2. Bivariate correlations with Hb level revealed a significant positive relationship with age (r = 0.150, P = 0.001), MUAC (r = 0.263, P < 0.001) and level of education (r = 0.145, P = 0.002) while a significant negative relationship was observed with MP status (r = -0.085, P = 0.045), type of school (r = -0.099, P = 0.024) and locality (r = -0.089, P = 0.038). In the MLR model, the only factors identified as significant predictors of Hb levels were the MUAC (P < 0.001) and the MP status (P = 0.035) as shown in Table 6.

**Table 4. Prevalence and severity of anaemia with respect to sex, age, locality, type of school and malaria parasite.**

| Characteristic | | N | Anaemia prevalence % (n) | χ² P value | Mild anaemia prevalence % (n) | Moderate anaemia prevalence % (n) | Severe anaemia prevalence % (n) | χ² P value |
|---|---|---|---|---|---|---|---|---|
| **Sex** | Male | 189 | 74.6 (141) | | 13.7 (29) | 61.3 (130) | 0.9 (2) | |
| | Female | 212 | 75.9 (161) | 0.756 | 12.2 (23) | 59.8 (113) | 2.6 (5) | 0.584 |
| **Age group in years** | ≤ 6 | 60 | 81.7 (49) | | 6.7 (4) | 75.0 (45) | 0.0 (0) | |
| | 7–10 | 234 | 74.8 (175) | 0.433 | 11.1 (26) | 61.5 (144) | 2.1 (5) | **0.034** |
| | 11–14 | 107 | 72.9 (78) | | 20.6 (22) | 50.5 (54) | 1.9 (2) | |
| **Locality** | Owe | 191 | 70.7 (135) | | 12.6 (24) | 55.5 (106) | 2.6 (5) | |
| | Meanja | 38 | 89.5 (34) | **0.035** | 13.2 (5) | 76.3 (29) | 0.0 (0) | 0.132 |
| | Mpundu | 172 | 77.3 (133) | | 13.4 (23) | 62.8 (108) | 1.2 (2) | |
| **School Type** | Catholic | 233 | 73.0 (170) | 0.199 | 14.2 (33) | 57.5 (134) | 1.3 (3) | 0.334 |
| | Government | 168 | 78.6 (131) | | 11.3 (19) | 64.9 (109) | 2.4 (4) | |
| **MP status** | Positive | 136 | 80.9 (110) | 0.064 | 13.2 (18) | 66.2 (90) | 1.5 (2) | 0.290 |
| | Negative | 265 | 72.5 (192) | | 12.8 (34) | 57.7 (153) | 1.9 (5) | |
| **MP density category** | Low | 85 | 81.6 (71) | | 17.2 (15) | 63.2 (55) | 1.1 (1) | |
| | Moderate | 40 | 80.0 (40) | 0.948 | 5.0 (2) | 72.5 (29) | 2.5 (1) | 0.669 |
| | High | 9 | 77.8 (9) | | 11.1 (1) | 66.7 (6) | 0.0 (0) | |

P—values in bold are statistically significant.

Mild anaemia = Hb between 10.0 g/dL and the level in the WHO reference values for age.

Moderate anaemia = Hb between 7.0–9.9 g/dL based on WHO reference for age.

Severe anaemia = Hb < 7 g/dL and WHO reference for age.

Low MP density = ≤ 500 parasite /µL of blood.

Moderate malaria parasite density = 501–5000 parasites/µL of blood.

High malaria parasite density = >5000 parasites/µL of blood.

### Health status in the population

Based on the Hb level (Hb > 11g/dL) and the absence of fever, MP, STH and malnutrition, 13.7% (95% CI = 10.7–17.4%) of the SAC were considered as healthy in the study population. The proportion of healthy children was comparable among the age groups (≤ 6 years = 13.3%, 7–10 years = 15% and 11–14 years = 11.2%; P = 0.645) and sex (female = 13.7%, male = 13.8%; P = 0.982). Among the 195 negative for MP, STH, and malnutrition, 55 (28.2%, 95% CI = 22.4–34.9%) were classified as healthy. Even though more males (32.1%) and children of the 11–14 years age group (33.3%) were healthier than females (25.4%) and those ≤ 6 years (24.2%) and 7–10 years (27.8%) old the differences were not statistically significant (P = 0.308, P = 0.693), respectively.

Among the 346 (86.3%) unhealthy children, anaemia occurred most frequently (40.5%), followed by MP (29.8%) and malnutrition (18.8%) as shown in Fig 3. The occurrence of the various conditions varied significantly ($χ^2$ = 34.88, P < 0.001) with the age group. Children in the ≤ 6 years group had the highest prevalence of anaemia (48.1%) and MP (38.5%) while those of the 11–14 years age group had the highest prevalence of malnutrition (25.3%), MP + malnutrition (15.8%), malnutrition + STH (2.1%) and MP + STH (3.2%) as revealed in Fig 3.

### Discussion

Understanding the burden of malaria, malnutrition and helminth infection in SAC is essential to finding delivery mechanisms to help implement control measures in this at-risk population. This study assessed the health status of SAC with respect to malaria, malnutrition and STH infections using the level of Hb as an indicator.

**Table 5.** Anaemia prevalence and mean (SD) Hb (g/dL) levels by sex and age as influenced by infection category and nutritional status.

| Parameter | n | Prevalence of anaemia % (n) | Overall mean (SD) Hb (g/dL) level | Mean (SD) Hb (g/dL) level by sex | | P value[a] | Mean (SD) Hb (g/dL) level by age group in years | | | P- Value[b] |
|---|---|---|---|---|---|---|---|---|---|---|
| | | | | Male | Female | | ≤ 6 | 7–10 | 11–14 | |
| **STH Only** | 3 | 66.7 (2) | 10.3 (1.8) | 10.0 (2.4) | 11.0 (-) | - | 8.3 | 11.4 (0.5) | - | - |
| **MP Only** | 103 | 79.6 (82) | 10.5 (1.2) | 10.4 (1.2) | 10.5 (1.1) | 0.706 | 10.0 (0.9)[c] | 10.4 (1.2) | 10.9 (1.3)[c] | **0.022** |
| **Malnutrition only** | 65 | 73.8 (48) | 10.5 (1.6) | 10.2 (1.7) | 10.8 (1.4) | 0.296 | 10.3 (0.7) | 10.4 (1.4) | 10.6 (20) | 0.753 |
| **MP + Malnutrition** | 29 | 86.2 (25) | 10.4 (1.2) | 10.5 (1.3) | 10.2 (1.1) | 0.326 | 9.0 (0.3) | 10.3 (1.1) | 10.6 (1.2) | 0.143 |
| **Negative** | 195 | 71.8 (140) | 10.8 (1.3) | 10.9 (1.3) | 10.7 (1.4) | 0.338 | 10.4 (1.3) | 10.8 (1.3) | 11.2 (1.4) | 0.058 |
| **Total** | 401 | 75.3 (302) | 10.6 (1.4) | 10.6 (1.3) | 10.6 (1.4) | 0.889 | 10.2 (1.2)[d] | 10.6 (1.3) | 10.9 (1.5)[d] | **0.004** |

Abbreviations: STH Soil transmitted helminth, MP Malaria parasite

MP Only = Infection with malaria parasite only.

Malnutrition only = Participants with malnutrition only (Absence of MP and STH).

MP + Malnutrition = Participants with malaria parasite and malnutrition.

Negative = Negative for MP and STH as well as absence of malnutrition.

P–value[a] obtained by t- test.

P–value[b] obtained by ANOVA.

Figures in bold are statistically significant.

[c] Mean (SD) Hb (g/dL) significantly different (Post Hoc Turkey HSD test: P = 0.017).

[d] Mean (SD) Hb (g/dL) significantly different (Post Hoc Turkey HSD test: P = 0.003).

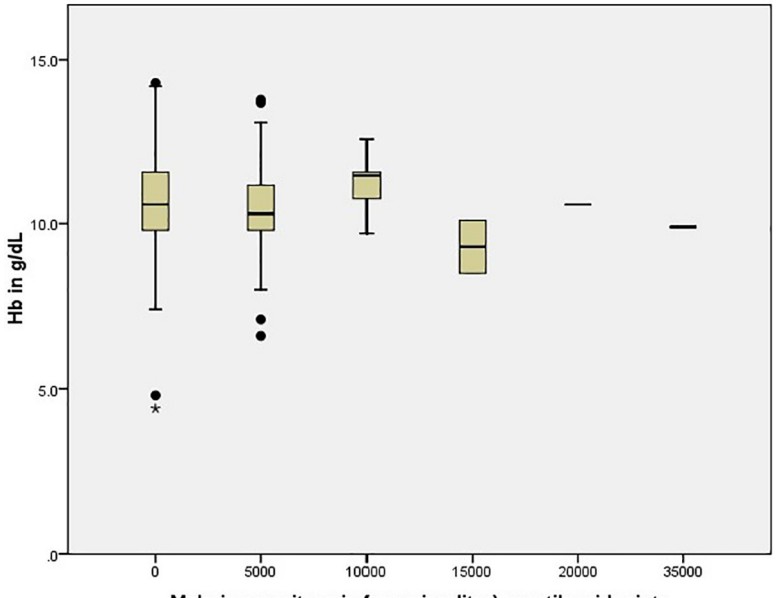

**Fig 2. Haemoglobin level by malaria parasitaemia in the study population.**

**Table 6. MLR model showing influence of independent variables on haemoglobin level.**

| Independent variable | B | Standard error | 95% CI | P- value |
|---|---|---|---|---|
| Age | -0.007 | 0.041 | -0.088–0.075 | 0.873 |
| Sex | 0.032 | 0.134 | -0.231–0.295 | 0.813 |
| MUAC | 0.157 | 0.042 | -0.074–0.240 | <0.001*** |
| Nutritional status | -0.113 | 0.166 | -0.440–0.214 | 0.498 |
| MP status | -0.294 | 0.139 | -0.567 - -0.021 | 0.035* |
| Level of education | 0.148 | 0.086 | -0.021–0.316 | 0.085 |
| Type of School | -0.204 | 0.135 | -0.468–0.061 | 0.131 |
| Locality | -0.066 | 0.070 | -0.206–0.072 | 0.347 |

Abbreviations: MUAC Mid upper arm circumference, MP Malaria parasite.

Dependent variable: Hb (g/dl)

* P is significant at the 0.05 level.

*** P is significant at the 0.001 level. Model summary: R = 0.311, $R^2$ = 0. 097, Adjusted $R^2$ = 0.078, F = 5.251, P < 0.01, N = 401.

The prevalence of STH infections was very low (2.2%) when compared with that (33.7%) of a similar study carried out in the same region two years earlier [32]. However, it is comparable to the 1% obtained in SAC in Tiko Health District [33] and 2.5% in selected rural, semi-urban and urban communities [22] all in the Mount Cameroon area. While a high burden of infection is not very common in regions targeted for elimination, the low prevalence could be as a result of the mass chemotherapy with mebendazole in school children initiated in Cameroon through the Ministry of Public Health since 2004 [5]. Furthermore, a combination of auto-medication, history of chemotherapy, changes in environmental and behavioural factors by individuals may have led to the sustained low prevalence of STH infection in the area.

The low occurrence of STH infection in SAC in the area limits the expression of its influence on the Hb level. However, out of the 3 children who were infected with STH only, 2 (66.7%) were anaemic. Moreover, those with STH infection had lower mean Hb levels when compared with their negative counterparts. Although this observation is limiting, it corroborates the negative effect of STH infection on the Hb values reported in abundant STH (*Ascaris lumbricoides, Trichuris trichiura*) infections among school children of Kashmir Valley India [34].

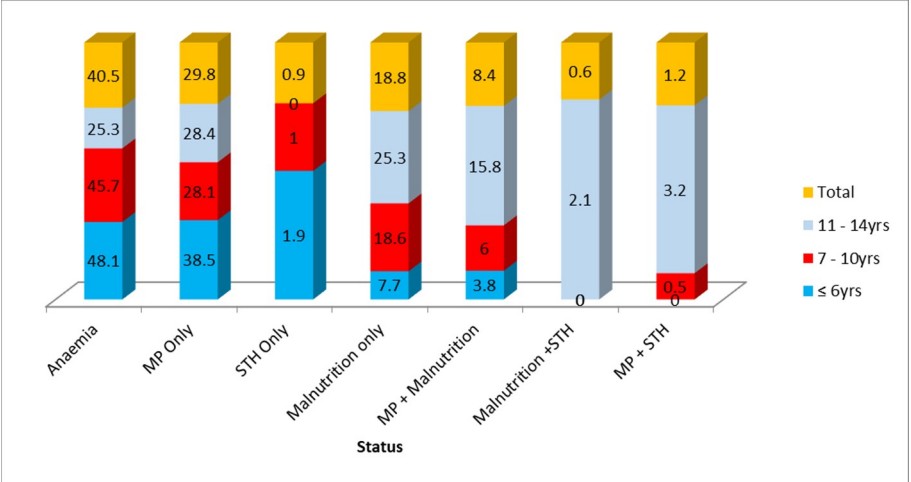

**Fig 3. Prevalence (%) of the different conditions in unhealthy participants as influenced by age.**

The overall MP prevalence (33.9%) is comparable to the 33.0% obtained in school children in Bomaka and Molyko in the Mount Cameroon Area [35] while, it is lower when compared with the 44.26% obtained by Kimbi *et al.* [36] in SAC children in Muea in the Mount Cameroon area and the 50.7% by Makoge *et al.* [37] in primary school pupils in Mbonge Sub-Division, Cameroon. *Plasmodium falciparum* infection prevalence as high as 60% in school children has been reported in Malawi [38] and a comparable prevalence of 30% in similar transmission settings in Uganda [39]. It is worth noting that majority (83.8%) of the MP infections were asymptomatic as most SAC do not have any symptoms because they have acquired some immunity. While acknowledging the decline in malaria prevalence due to sustained malaria control interventions implemented by the Cameroon Government through the National Malaria Control Programme [40] this burden highlights the need to embark upon malaria control in SAC in Cameroon.

Although all three localities are rural, children of Meanja had the highest significant prevalence of MP. More specifically the prevalence was highest in SAC of the 11–14 years age group when compared with the other age groups. As expected, the heterogeneity in prevalence of malaria parasite is demonstrated in the area of study. Even though the distribution of malaria cases with age is influenced by the transmission intensity [41], it is likely that other ecological factors and behaviours favoured the occurrence and age-related distribution of the infection in the area. Inhabitants of Meanja live near the CDC rubber and palm plantations which together with a lot of bushes around, creates an ecosystem appropriate for *Anopheles* mosquito development in the locality. In addition, older children in rural areas are more involved in outdoor activities such as farming and household activities like fetching water from far water sources early in the morning exposing them to the mosquito vector.

The high prevalence of anaemia (75.3%) in SAC in this rural area with a significant majority (60.6%) being moderate anaemia is an indication that anaemia is a major public health problem. This prevalence was higher than those obtained by other authors in the region and elsewhere [22, 37, 38, 42]. This probably reflects the poor state of health of children in the area. The aetiology of anaemia in SAC is multifactorial and the relative importance of each cause varies from place to place. Albeit it is difficult to differentiate the influence of malaria parasite on anaemia in SAC, the attributable risk of anaemia due to malaria parasite (3.8%), malnutrition only (1.1%) and MP and malnutrition (0.96%) is very low. However, it is very evident from the findings that SAC with these conditions especially those with MP and malnutrition had the highest prevalence of anaemia and lowest Hb levels when compared with their negative counterparts suggesting their contributions to the burden of anaemia in the area. The highest occurrence of anaemia in SAC of Meanja locality (89.5%) that also had the highest prevalence of malaria parasite lends more support to this assertion.

The observed prevalence of malnutrition (24.4%) in SAC in this area shows it is a public health issue. The very low mean HA (-1.17) and WA (-0.33) z scores culminating in the presence of stunting (23.7%) and underweight (4.5%) observed in the children highlights the degree of growth failure in height and weight in SAC in this rural area. The prevalence of stunting is similar to that observed in primary school children in Nairobi-Kenya [43] but higher than the 11.3% observed in SAC in North-Eastern Ethiopia [44]. Male children like their counterparts in Kenya and Ethiopia [44, 45] had a significant tendency of being stunted than females while a study in Southern Ethiopia [46] showed no major differences in prevalence of stunting in males and females. The growth and development of male children is influenced by environmental and nutritional stress more than the female [47]. Hence, it is likely that the mean energy intake for boys in this rural area did not meet the energy requirements as boys are more hyperactive in this age range than females who spend more time in food preparation and may thus have increasingly access to excess food.

In line with the other studies stunting was found to be significantly higher in the older age group [44, 46]. Stunting which is a marker of chronic malnutrition is more likely to be apparent with increase in age. Hence, its increased presence in older children which in addition are in a transition to the adolescent stage in life that has its own nutritional needs [48] which may not be adequate in this rural environment.

Findings from the study revealed the mean Hb level of SAC in these rural settings are comparatively lower than the WHO level for age and sex even though an ideal Hb level is not yet established. While the mean Hb level in all conditions were lower, that of children in the 6 years age group and below with malaria parasite infection was statistically significant. Furthermore, the model identified MP status as a significant predictor of Hb level with a negative relationship. Several other studies have associated *Plasmodium* infection to lower Hb level [42, 49, 50]. This highlights the insidious effect asymptomatic malaria parasitaemia may have on the Hb level of younger children who have not developed anti disease immunity.

On the other hand, the MUAC was identified as a significant predictor of Hb level with a positive association as shown in the MLR model. While further investigation is necessary to assert this association, the MUAC which measures only acute malnutrition could be an alternative of great value especially in resource poor settings were Hb measurement is unaffordable. Blood Hb measurement, a common indicator for diagnosing anaemia, requires trained personnel, expensive equipment or well-developed laboratory facilities which is often unavailable in rural areas. Measuring the MUAC is much cheaper and easier than measuring weight and height. In addition, it is less affected by acute dehydration than weight-based indices [51].

Furthermore, keeping an update information on the level of education of parent or caregiver is of utmost value as findings in a bivariate analysis demonstrated a significant positive association of parent/caregiver level of education with the Hb level. While acknowledging the limitations of self-reported levels of education and its potential of introducing bias, this finding is not surprising as low educational level can lead to low income and socio-economic status and thus inability to provide for proper feeding and affordable health care. Basic causes of anaemia reported in other studies include, maternal level of education and household wealth rank [52].

The magnitude of unhealthy SAC (86.3%) in the rural setting surpasses the occurrence of healthy children and that is a cause for concern. Although the presence of these conditions contributed to the unhealthy status of SAC, the fact that anaemia occurred in 71.8% of the negatives demonstrates their health is compromised by other ailments that were not investigated in the study. One of the limitations of this study is that the number of infections investigated are fewer albeit those evaluated are reported to be of common occurrence in the Mount Cameroon area [8, 37]. More research involving a wider range of morbidities (micro-nutrient deficiency, bacterial and viral infections) other than the conditions investigated that may be responsible for the unhealthy status of the children needs to be carried out. However, the lowered haemoglobin level observed in measurable and unmeasurable ailments lends support to the fact that haemoglobin level could be used as an indicator of the health status of children.

## Conclusions

It is evident from this study that the health of majority of SAC in these rural settings in the Mount Cameroon area is compromised by malaria, helminthiasis, malnutrition and other conditions not investigated. Anaemia is of major public health concern and the Hb level could serve as a prognostic marker of the health status of SAC. Children with MP and malnutrition had the highest prevalence of anaemia and lowest Hb levels indicating their contributions to the burden of anaemia even though the attributable risk to it was insignificant. Malaria parasite

is a significant negative predictor of Hb level hence, there is a need for intervention pro-grammes targeting SAC in rural areas that integrates proper malaria control measures with improvement of educational level of parent/caregiver especially on proper nutrition and health care practices that will ensure health and well-being of the children.

## Supporting information

**S1 File. Questionnaire.**
(PDF)

## Acknowledgments

The authors are thankful to the village chiefs of Owe, Mpundu and Meanja, head teachers of the various primary schools, parents/guardians as well as the children who participated in the study. We acknowledge the support of IIE-SRF (Institute of International Education- Scholar Rescue Fund) and MPH programme, College of Veterinary Medicine, Cornell University, Ith-aca, New York in providing the fellowship and right academic environment respectively for the drafting of this manuscript.

## Author Contributions

**Conceptualization:** Irene Ule Ngole Sumbele.

**Data curation:** Irene Ule Ngole Sumbele, Ayeah Joy Nkain, Teh Rene Ning.

**Formal analysis:** Irene Ule Ngole Sumbele.

**Investigation:** Irene Ule Ngole Sumbele, Ayeah Joy Nkain, Teh Rene Ning, Judith Kuoh Anchang-Kimbi, Helen Kuokuo Kimbi.

**Methodology:** Irene Ule Ngole Sumbele, Ayeah Joy Nkain.

**Supervision:** Irene Ule Ngole Sumbele, Judith Kuoh Anchang-Kimbi.

**Validation:** Irene Ule Ngole Sumbele.

**Writing – original draft:** Irene Ule Ngole Sumbele.

**Writing – review & editing:** Irene Ule Ngole Sumbele, Helen Kuokuo Kimbi.

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
