## [Decision Letter · Decision Letter 0]

27 Nov 2019

PONE-D-19-31286

Haemoglobin level as an indicator of health status of school-aged children in Muyuka, Southwest Cameroon: Influence of malaria parasites, soil-transmitted helminths and malnutrition

PLOS ONE

Dear Dr. Sumbele,

Thank you for submitting your manuscript to PLOS ONE. After careful consideration, we feel that it has merit but does not fully meet PLOS ONE’s publication criteria as it currently stands. Therefore, we invite you to submit a revised version of the manuscript that addresses the points raised during the review process.

We would appreciate receiving your revised manuscript by Jan 11 2020 11:59PM. To enhance the reproducibility of your results, we recommend that if applicable you deposit your laboratory protocols in protocols.io, where a protocol can be assigned its own identifier (DOI) such that it can be cited independently in the future. For instructions see: http://journals.plos.org/plosone/s/submission-guidelines#loc-laboratory-protocols

We look forward to receiving your revised manuscript.

Kind regards,

Hesham

Hesham M. Al-Mekhlafi, PhD

Academic Editor

PLOS ONE

Journal Requirements:

1. Please ensure that your manuscript meets PLOS ONE's style requirements, including those for file naming. The PLOS ONE style templates can be found athttp://www.journals.plos.org/plosone/s/file?id=wjVg/PLOSOne_formatting_sample_main_body.pdf and http://www.journals.plos.org/plosone/s/file?id=ba62/PLOSOne_formatting_sample_title_authors_affiliations.pdf

2. Please address the following:

- Please refer to any post-hoc corrections to correct for multiple comparisons during your statistical analyses. If these were not performed please justify the reasons. Please refer to our statistical reporting guidelines for assistance (https://journals.plos.org/plosone/s/submission-guidelines.#loc-statistical-reporting).

- Please ensure you have thoroughly discussed any potential limitations of this study within the Discussion section, for example the potential bias introduced by using self-reported data.

- Please include additional information regarding the survey or questionnaire used in the study and ensure that you have provided sufficient details that others could replicate the analyses. For instance, if you developed a questionnaire as part of this study and it is not under a copyright more restrictive than CC-BY, please include a copy, in both the original language and English, as Supporting Information. In addition, please include any details of the pre-testing of this questionnaire.

Thank you for your attention to our queries.

3. Your ethics statement must appear in the Methods section of your manuscript. If your ethics statement is written in any section besides the Methods, please move it to the Methods section and delete it from any other section. Please also ensure that your ethics statement is included in your manuscript, as the ethics section of your online submission will not be published alongside your manuscript.

Additional Editor Comments (if provided):

Reviewers' comments:

Reviewer's Responses to Questions

**Comments to the Author**

1. Is the manuscript technically sound, and do the data support the conclusions?

Reviewer #1: Yes

Reviewer #2: No

Reviewer #3: Yes

2. Has the statistical analysis been performed appropriately and rigorously? 

Reviewer #1: Yes

Reviewer #2: No

Reviewer #3: Yes

3. Have the authors made all data underlying the findings in their manuscript fully available?

Reviewer #1: Yes

Reviewer #2: Yes

Reviewer #3: Yes

4. Is the manuscript presented in an intelligible fashion and written in standard English?

Reviewer #1: Yes

Reviewer #2: No

Reviewer #3: Yes

5. Review Comments to the Author

Reviewer #1: I read the manuscript with great interest. The manuscript was very interesting and provided good information on the Hemoglobin level as an indicator of health status of school-aged children. However, it has some points that need to be addressed to improve its quality and impact. Here are some of my observations:

The map for the target area should be included in the manuscript.

The references were mostly from more than 5 years ago. If possible, add more recent references.

Reviewer #2: Dear Editor

I have thoroughly read the manuscript, although it represents an important public health problem in School Aged Children it has a number of issues in literature review, statistical analysis and discussion. The jauthors should revise the manuscript by addressing a number of typographical, data presentation and discussion before it is considered for publication. I have highlighted a number of issues in the attached revised doc

Reviewer #3: Review for Manuscript Entitled “Haemoglobin level as an indicator of health status of school-aged children in Muyuka, Southwest Cameroon: Influence of malaria parasites, soil-transmitted helminths and malnutrition”

Abstract

Background

• Line 60- You mention an abbreviation SAC without prior statement of its long form. Much as it is appearing for the first time in the background section it has to be in its long form, and then can subsequently be mentioned as a acronym.

• Line 98- Put a percent symbol after 64.3

• Line 99-Put a percent symbol after 19.8

• Line 100-Indicate the specific threshold at which anaemia is considered a public health problem

Methodology

Study site

The description of the study site need to have a reference, no reference is provided for the description of the study site, kindly provide the reference.

Study design

• After stating the study design, for the purpose of making the paper easy to follow by the reader, this section should have the following subsections:

• Study population

• Sample size

• Sampling procedure

• In this paper, the primary objective was to determine the prevalence of Malaria parasite, STH and Malnutrition. But in calculating the sample size you used 14.0% which was the prevalence of STH, but you have not indicated that this is what is going to give enough sample size to estimate the population prevalence of Malaria parasites and Malnutrition as well.

• Line 146-148: You state that “The minimum sample size was calculated using the prevalence of P. falciparum malaria and helminth infections of previous studies: 35.5 and 14.0%, respectively, in the Mount Cameroon area”. But the formular for sample size calculation that you have provided allows for using only one proportion, how did you use these two proportions?

• But after you have calculated the sample size, you have not indicated in this section what was the calculated minimum sample size.

• Line 152-154: “Samples collected comprised of finger prick blood and stool for MP detection and speciation, Hb 153 measurement for assessment of anaemia and detection of egg or larva of STHs respectively” , rephrase this statement to make it clear as to which sample was used for whic purpose.

Clinical evaluation

• Line 172: You have written o.1cm instead of 0.1, kindly change accordingly.

Collection of blood and stool samples

• Line 192: Samples were fixed in 10% formalin and then examined using the Kato-katz technique, can the author explain, how did they perform kato-katz on a formalin fixed stool, Kato-katz performs well when done on fresh stool. Can you explain this?

• Line 194: You have written Kato-katz concentration technique. Kato-katz is not one of the concentration techniques applied on stool samples.

Statistical analysis

Line 228: SPSS is no longer Statistical Package for Social Sciences, please supply the correct long form.

Results

Characteristics of the study participants

• In this subsection you were supposed to provide only socio-demographic and economic profile of the study participants. But you have also provided findings from laboratory analysis and other measurements, these should fall under a separate subheading and a separate table.

• Line 260-261: Out of 9 children with STH, 55.5% were infected with Trichuris trichiura, please indicate both the number and percentages.

Anaemia prevalence and its severity

• When writing confidence interval, you must indicate that it is 95% CI, do not just write “CI” as it appears in line 293, 301, 302 etc.

6. PLOS authors have the option to publish the peer review history of their article (what does this mean?). If published, this will include your full peer review and any attached files.

Reviewer #1: Yes: Associate Professor Dr. Talal Alharazi

Reviewer #2: No

Reviewer #3: No

---

## [Author Response · Author response to Decision Letter 0]

31 Dec 2019

Response to Comments and Concerns of the Editor

1. Please ensure that your manuscript meets PLOS ONE's style requirements, including those for file naming. The PLOS ONE style templates can be found athttp://www.journals.plos.org/plosone/s/file?id=wjVg/PLOSOne_formatting_sample_main_body.pdf and http://www.journals.plos.org/plosone/s/file?id=ba62/PLOSOne_formatting_sample_title_authors_affiliations.pdf

The corrections in the naming of the files and author affiliations have been effected.

2. Please address the following:

- Please refer to any post-hoc corrections to correct for multiple comparisons during your statistical analyses. If these were not performed please justify the reasons. Please refer to our statistical reporting guidelines for assistance (https://journals.plos.org/plosone/s/submission-guidelines.#loc-statistical-reporting).

The Post Hoc Turkey HSD test has been included where significant differences in means were observed as indicated. The correction is highlighted in Turquoise.

- Please ensure you have thoroughly discussed any potential limitations of this study within the Discussion section, for example the potential bias introduced by using self-reported data.

The factors discussed where measurable parameters that limited the potential of introducing a bias. However, limitations in the study have been reported in lines 501-503 and the addition of potential limitations highlighted as indicated.

- Please include additional information regarding the survey or questionnaire used in the study and ensure that you have provided sufficient details that others could replicate the analyses. For instance, if you developed a questionnaire as part of this study and it is not under a copyright more restrictive than CC-BY, please include a copy, in both the original language and English, as Supporting Information. In addition, please include any details of the pre-testing of this questionnaire.

The simple pre-tested questionnaire used has been included as supporting information 1.

Thank you for your attention to our queries.

3. Your ethics statement must appear in the Methods section of your manuscript. If your ethics statement is written in any section besides the Methods, please move it to the Methods section and delete it from any other section. Please also ensure that your ethics statement is included in your manuscript, as the ethics section of your online submission will not be published alongside your manuscript.

The ethics statement has been moved to the methods section as requested

Reviewer comments and responses

The changes made in the manuscript based on reviewers’ comments and concerns have been highlighted in yellow and green in the manuscript with tract changes for easy appraisal.

Reviewer #1: I read the manuscript with great interest. The manuscript was very interesting and provided good information on the Hemoglobin level as an indicator of health status of school-aged children. However, it has some points that need to be addressed to improve its quality and impact. Here are some of my observations:

The map for the target area should be included in the manuscript.

The map of the study area has been included in the manuscript as requested see (Fig 1).

The references were mostly from more than 5 years ago. If possible, add more recent references.

While a single reference or two has been added, the authors believe they did an appropriate citation of relevant and recent articles in the field. At least 20 of the articles cited were published between 2014 and 2019. The articles beyond that cited are very relevant to the background methods and discussion of the results. 

Reviewer #2: Dear Editor

I have thoroughly read the manuscript, although it represents an important public health problem in School Aged Children it has a number of issues in literature review, statistical analysis and discussion. The jauthors should revise the manuscript by addressing a number of typographical, data presentation and discussion before it is considered for publication. I have highlighted a number of issues in the attached revised doc

The corrections effected as requested by the reviewer are highlighted in yellow.

• Line 60: The correction has been effected.

• Line 62: The correction has been effected.

• Line 92: Literature has been reviewed and a sentence added thus “Manifestation of malnutrition is often observed in terms of anaemia, micronutrient deficiencies (iron, folic acid, vitamin B12) and anthropometric measurements.”

• Line 96: The aetiology of anaemia has been included as requested.

• Line 98: The sentence has been rephrased for clarity.

• Line 99: The prevalence of anaemia stated in line 103, specifically for SAC is the most recent. The updates given by WHO in 2011 does not include this specific age group. The recent updates are for children 6 months -5 years, pregnant women and women between the ages of 14 and 49 of various categories.

• The sentences in line 99-104 have been revised for clarity as requested.

• Line 140: The number of schools from which selection was made had been included and the method of selection as follows “Out of a list of 9 schools operating officially. Random selection by balloting was used to obtain a representative sample”.

• Line 151: The authors believe using the prevalence of malaria parasite and STH to estimate the population size to be sampled is appropriate. The main aim of this paper is to assess the influence of these infections on the haemoglobin level of SAC. Anaemia is one of the outcomes of these infections. Infections may not necessarily lead to anaemia in every circumstance.

• Line 153: Note taken. Correction effected.

• Line 153: The laboratory method has been included.

• Line 162: The authors believe the ensuing statement clarifies the previous sentence. However, the questionnaire has been attached as a supporting information for clarity (S1 Questionnaire).

• Line 172: Correction effected.

• Line 185: Haemoglobin measurement and anaemia classification has been combined as requested.

• Line 223: The time (When?)“within 24 hours” has been included.

• Line 230: Percentages has been replaced with proportions as requested.

• Line 249: The correction has been effected as indicated.

• Table 1: It is actually ITN Use and not possession. The correction has been effected based on the question in the questionnaire and the responses of the participants.

• Lines 271-273: The sentence has been rephrased for clarity as requested.

• Table 2: The phrase “as influenced by” has been deleted as requested.

• Lines 294-296 has been rephrased for clarity as requested.

• Line 300: The Hb range of moderate anaemia has been included as requested.

• Line 303. The “with” has been replaced with “by” as requested.

• Table 3: The definitions of the various categories of anaemia has been provided again beneath the table as end note even though it has been clearly defined in the methods.

• Line 314: The title has been rephrased as requested.

• Line 315: The correction has been effected.

• Line 380: Correction has been effected.

• Lines 392-394: The sentence has been rephrased for clarity. Secondly the sentence does not need any reference because the possible explanation given for the low prevalence of STH observed in the area is based on the responses obtained from the questionnaire.

• Line 398: The sentence has been rephrased for clarity.

• Conclusion: The statement has been revised.

Reviewer #3: Review for Manuscript Entitled “Haemoglobin level as an indicator of health status of school-aged children in Muyuka, Southwest Cameroon: Influence of malaria parasites, soil-transmitted helminths and malnutrition” 

Corrections of Reviewer 3 are highlighted in green

Abstract

Background

• Line 60- You mention an abbreviation SAC without prior statement of its long form. Much as it is appearing for the first time in the background section it has to be in its long form, and then can subsequently be mentioned as a acronym.

The abbreviation has been written in full as indicated

• Line 98- Put a percent symbol after 64.3

The symbol has been added.

• Line 99-Put a percent symbol after 19.8

The symbol has been included.

• Line 100-Indicate the specific threshold at which anaemia is considered a public health problem

Anaemia is considered a public health problem when the prevalence is ≥ 5%. That has been included.

Methodology

Study site

The description of the study site need to have a reference, no reference is provided for the description of the study site, kindly provide the reference.

The reference has been included.

Study design

• After stating the study design, for the purpose of making the paper easy to follow by the reader, this section should have the following subsections:

• Study population

• Sample size

• Sampling procedure

The study design has been organized in the different subsections as requested

• In this paper, the primary objective was to determine the prevalence of Malaria parasite, STH and Malnutrition. But in calculating the sample size you used 14.0% which was the prevalence of STH, but you have not indicated that this is what is going to give enough sample size to estimate the population prevalence of Malaria parasites and Malnutrition as well.

The prevalence of both MP and helminth were used in determining the minimum sample size. The correction has been effected as indicated in the highlight.

• Line 146-148: You state that “The minimum sample size was calculated using the prevalence of P. falciparum malaria and helminth infections of previous studies: 35.5 and 14.0%, respectively, in the Mount Cameroon area”. But the formular for sample size calculation that you have provided allows for using only one proportion, how did you use these two proportions?

Yes the formula indicates just one prevalence. Hence, the sample size was calculated twice using both prevalence and an average was obtained. That has been indicated in the corrected version of the manuscript.

• But after you have calculated the sample size, you have not indicated in this section what was the calculated minimum sample size.

A minimum sample size of 268 samples obtained has been included.

• Line 152-154: “Samples collected comprised of finger prick blood and stool for MP detection and speciation, Hb 153 measurement for assessment of anaemia and detection of egg or larva of STHs respectively” , rephrase this statement to make it clear as to which sample was used for whic purpose.

The sentence has been rephrased for clarity as requested.

Clinical evaluation

• Line 172: You have written o.1cm instead of 0.1, kindly change accordingly.

The correction has been effected.

Collection of blood and stool samples

• Line 192: Samples were fixed in 10% formalin and then examined using the Kato-katz technique, can the author explain, how did they perform kato-katz on a formalin fixed stool, Kato-katz performs well when done on fresh stool. Can you explain this?

 The statement has been rephrased for clarity. The stool samples were not fixed but in order to maintain the morphology of the egg during the period of transportation before examination in the laboratory, the samples were preserved in 10% formalin. This has been reported to preserve the morphology of Hookworm for up to 15 days and Trichuris for up to 30 days. We actually went ahead to compare our observations with that of formol ether concentration techniques even though that was not reported here because that is not the subject of this paper.

• Line 194: You have written Kato-katz concentration technique. Kato-katz is not one of the concentration techniques applied on stool samples.

The word concentration has been deleted from the sentence. The authors are sorry for the mix up. It is due to the fact that we carried out both a concentration method (formol ether) which we didn’t report here and the Kato-katz technique.

Statistical analysis

Line 228: SPSS is no longer Statistical Package for Social Sciences, please supply the correct long form.

The name has been corrected to IBM-SPSS.

Results

Characteristics of the study participants

• In this subsection you were supposed to provide only socio-demographic and economic profile of the study participants. But you have also provided findings from laboratory analysis and other measurements, these should fall under a separate subheading and a separate table.

The laboratory analysis and other measurements have been separated from socio-demographic characteristic as requested.

• Line 260-261: Out of 9 children with STH, 55.5% were infected with Trichuris trichiura, please indicate both the number and percentages.

The correction has been effected.

Anaemia prevalence and its severity

• When writing confidence interval, you must indicate that it is 95% CI, do not just write “CI” as it appears in line 293, 301, 302 etc.

The corrections have been effected as indicated.

---

## [Decision Letter · Decision Letter 1]

29 Jan 2020

PONE-D-19-31286R1

Haemoglobin level as an indicator of health status of school-aged children in Muyuka, Southwest Cameroon: Influence of malaria parasites, soil-transmitted helminths and malnutrition

PLOS ONE

Dear Dr. Sumbele,

Thank you for submitting your manuscript to PLOS ONE. After careful consideration, we feel that it has merit but does not fully meet PLOS ONE’s publication criteria as it currently stands. Therefore, we invite you to submit a revised version of the manuscript that addresses the points raised during the review process.

We would appreciate receiving your revised manuscript by Mar 14 2020 11:59PM. To enhance the reproducibility of your results, we recommend that if applicable you deposit your laboratory protocols in protocols.io, where a protocol can be assigned its own identifier (DOI) such that it can be cited independently in the future. For instructions see: http://journals.plos.org/plosone/s/submission-guidelines#loc-laboratory-protocols

We look forward to receiving your revised manuscript.

Kind regards,

Hesham

Hesham M. Al-Mekhlafi, PhD

Academic Editor

PLOS ONE

Additional Editor Comments (if provided):

Academic Editor’s Comments:

Dear authors,

Thank you for submitting your revised manuscript to PLoS One. This version has addressed the reviewers’ comments; however, corrections remain and additional comments should be addressed before this manuscript can be accepted. Please consider the following:

1- In laboratory procedures section, it is mentioned that malaria parasite density per μL of blood was determined; however, information on this variable was not provided in the manuscript. Using the term “malaria parasitaemia” in the current manuscript without providing these results is unclear and can be misleading. It can be replaced with malaria infection. Moreover, it is interesting to examine the association between the level of parasitaemia (parasite density) and Hb level and anaemia prevalence as well, and this should be incorporated into the results & discussion of this manuscript.

2- The attributable risk (AR%) was mentioned in statistical analysis section (please remove details on formula used) and in discussion and conclusion; however, results were not provided in results section.!

3- I suggest change the title. Presentation of your results and your conclusion are not compatible with the title “Haemoglobin level as an indicator of health status of ….”. Basically, the title can be “Influence of malaria, soil-transmitted helminth infections and malnutrition on haemoglobin level and anaemia prevalence among school-aged children in Muyuka, Southwest Cameroon”. You may think about alternative titles.

4- Tables format should be improved and edited for consistency and English. For instance:

4.1. Table 5: change columns of total to “overall mean (SD) Hb level” and this column can be placed after prevalence o anaemia column. P-value for sex can be placed after the columns of mean Hb for sex and p-value for age groups will be accordingly after the columns of age. Indicate the use of t-test and ANOVA in the footnotes.

4.2. Table 3: the data are % (n) but it is indicated as (n) only.

4.3. Table 6: please report important necessary results only for the output of MLR analysis; collinearity statistics and correlation coefficient columns can be removed. Please read about reporting such results in scientific papers or refer to previous literature. For instance, provide B value, standard error, 95% CI and p value for the included variables.

5- Figure 2 can be removed. Renumber the figures.

6- What is URIT-12 haemoglobin meter? Company, City, Country, and cite reference for its validity.

7- Follow first appearance rule; for instance, spell out the MUAC in abstract and then add the abbreviation between brackets. Follow this in the entire text and tables. In tables, define the abbreviations in the footnotes.

8- Moreover, the manuscript needs extensive editing by a native English ‎speaker. Please ‎note that poor English may ‎ultimately be a reason to reject the ‎manuscript.‎ ‎

Reviewers' comments:

Reviewer's Responses to Questions

**Comments to the Author**

1. If the authors have adequately addressed your comments raised in a previous round of review and you feel that this manuscript is now acceptable for publication, you may indicate that here to bypass the “Comments to the Author” section, enter your conflict of interest statement in the “Confidential to Editor” section, and submit your "Accept" recommendation.

Reviewer #2: All comments have been addressed

Reviewer #3: All comments have been addressed

2. Is the manuscript technically sound, and do the data support the conclusions?

Reviewer #2: Yes

Reviewer #3: Yes

3. Has the statistical analysis been performed appropriately and rigorously? 

Reviewer #2: Yes

Reviewer #3: Yes

4. Have the authors made all data underlying the findings in their manuscript fully available?

Reviewer #2: Yes

Reviewer #3: Yes

5. Is the manuscript presented in an intelligible fashion and written in standard English?

Reviewer #2: No

Reviewer #3: Yes

6. Review Comments to the Author

Reviewer #2: The authors have adequately addressed all my comments. However, proof reading of the manuscript is needed before publication

Reviewer #3: (No Response)

7. PLOS authors have the option to publish the peer review history of their article (what does this mean?). If published, this will include your full peer review and any attached files.

Reviewer #2: No

Reviewer #3: No

---

## [Author Response · Author response to Decision Letter 1]

12 Feb 2020

Response to Comments and Concerns of the Editor

Academic Editor’s Comments:

Dear authors,

Thank you for submitting your revised manuscript to PLoS One. This version has addressed the reviewers’ comments; however, corrections remain and additional comments should be addressed before this manuscript can be accepted. Please consider the following:

1- In laboratory procedures section, it is mentioned that malaria parasite density per μL of blood was determined; however, information on this variable was not provided in the manuscript. Using the term “malaria parasitaemia” in the current manuscript without providing these results is unclear and can be misleading. It can be replaced with malaria infection. Moreover, it is interesting to examine the association between the level of parasitaemia (parasite density) and Hb level and anaemia prevalence as well, and this should be incorporated into the results & discussion of this manuscript.

The authors apologize for not detailing the report of the malaria parasite (MP) density in the manuscript. The relationship between MP density and anaemia was reported in Table 4 earlier but the categories were not clearly defined as shown in the highlight. We have included the definition of low, moderate and high parasite densities in the method and as a foot note in the Table for clarity. As concerns malaria parasitaemia, the mean MP density and range has been included in Table 2 and the relationship between MP and Hb level reported in Fig 2. No statistically significant association was observed between MP density and anemia likewise between Hb level and MP even though a seemingly decreasing trend was revealed in Figure 2. This was taken into consideration during the discussion earlier therefore, the discussion was based on the prevalence of MP and AR in association with the other conditions rather than density alone. Hence, the authors don’t see the need for more discussion on malaria parasite density and Hb level in this instance. Furthermore, the relationship between Hb level and malaria parasitaemia was discussed further in Lines 508-511.

2- The attributable risk (AR%) was mentioned in statistical analysis section (please remove details on formula used) and in discussion and conclusion; however, results were not provided in results section.!

The authors wish to disagree on this point. The results on attributable risk were presented in lines 349 to 350 in the results as indicated in the highlight that is why they were discussed. In the context of the study, this calculation is necessary to show the relative contribution of each factor to the occurrence of anaemia.

3- I suggest change the title. Presentation of your results and your conclusion are not compatible with the title “Haemoglobin level as an indicator of health status of ….”. Basically, the title can be “Influence of malaria, soil-transmitted helminth infections and malnutrition on haemoglobin level and anaemia prevalence among school-aged children in Muyuka, Southwest Cameroon”. You may think about alternative titles.

The title has been revised to “Influence of malaria, soil-transmitted helminth infections and malnutrition on haemoglobin level among school-aged children in Muyuka, Southwest Cameroon: a cross sectional study on outcomes. We the authors believe this is more embracing. Placing haemoglobin level and anemia prevalence on the title is kind of odd. Anaemia is an outcome of these infections. It as well embraces the health status.

4- Tables format should be improved and edited for consistency and English. For instance:

The tables have been edited and formatted as shown in the tracked changes.

4.1. Table 5: change columns of total to “overall mean (SD) Hb level” and this column can be placed after prevalence o anaemia column. P-value for sex can be placed after the columns of mean Hb for sex and p-value for age groups will be accordingly after the columns of age. Indicate the use of t-test and ANOVA in the footnotes.

The corrections have been effected as requested. See tract changes for the corrections.

4.2. Table 3: the data are % (n) but it is indicated as (n) only.

Since the table is reporting prevalence, we the authors thought it wasn’t necessary to indicate percentage again as the unit of measurement. However, we have included it as requested. See tracked changes.

4.3. Table 6: please report important necessary results only for the output of MLR analysis; collinearity statistics and correlation coefficient columns can be removed. Please read about reporting such results in scientific papers or refer to previous literature. For instance, provide B value, standard error, 95% CI and p value for the included variables.

The Table was a composite table of two different out puts. However, the table has been revised as requested to include the B value, standard error and 95% CI for variables calculated. 

5- Figure 2 can be removed. Renumber the figures.

Figure 2 has been deleted and the results written out for clarity. A new Fig 2. that shows the Hb level and malaria parasite counts in the population has been produced in response to editors request of evidence of parasitaemia in the manuscript.

6- What is URIT-12 haemoglobin meter? Company, City, Country, and cite reference for its validity.

The specification has been included “URIT Medical Electronic co., Ltd, London, United Kingdom)”.

7- Follow first appearance rule; for instance, spell out the MUAC in abstract and then add the abbreviation between brackets. Follow this in the entire text and tables. In tables, define the abbreviations in the footnotes.

The corrections have been effected as indicated in the tracked changes.

8- Moreover, the manuscript needs extensive editing by a native English ‎speaker. Please ‎note that poor English may ‎ultimately be a reason to reject the ‎manuscript.

The manuscript has been extensively edited by a native English speaker as requested. See the tracked changes for the copy editing.‎ ‎

Reviewers' comments 

The comments in the file attached by reviewer 2 had been adequately addressed in the first round of review process as indicated by the reviewer.

---

## [Editor Report · Decision Letter 2]

11 Mar 2020

Influence of malaria, soil-transmitted helminths and malnutrition on haemoglobin level among school-aged children in Muyuka, Southwest Cameroon: a cross-sectional study on outcomes

PONE-D-19-31286R2

Dear Dr. Sumbele,

We are pleased to inform you that your manuscript has been judged scientifically suitable for publication and will be formally accepted for publication once it complies with all outstanding technical requirements.

With kind regards,

Hesham

Hesham M. Al-Mekhlafi, PhD

Academic Editor

PLOS ONE

Additional Editor Comments (optional):

Please note that weight and height values (mean and range) in Table 2 are in opposite order. To correct this, variable names should be exchanged. This can be corrected prior to the production stage.
---

## [Editor Report · Acceptance letter]

16 Mar 2020

PONE-D-19-31286R2 

Influence of malaria, soil-transmitted helminths and malnutrition on haemoglobin level among school-aged children in Muyuka, Southwest Cameroon: a cross-sectional study on outcomes 

Dear Dr. Sumbele:

I am pleased to inform you that your manuscript has been deemed suitable for publication in PLOS ONE. Congratulations! Your manuscript is now with our production department. 

With kind regards,

on behalf of

Dr. Hesham M. Al-Mekhlafi 

Academic Editor

PLOS ONE